# Design and Nonadiabatic Photoisomerization Dynamics Study of a Three-Stroke Light-Driven Molecular Rotary Motor

**DOI:** 10.3390/ijms23073908

**Published:** 2022-03-31

**Authors:** Jianzheng Ma, Sujie Yang, Di Zhao, Chenwei Jiang, Zhenggang Lan, Fuli Li

**Affiliations:** 1Ministry of Education Key Laboratory for Nonequilibrium Synthesis and Modulation of Condensed Matter, Shaanxi Province Key Laboratory of Quantum Information and Quantum Optoelectronic Devices, School of Physics, Xi’an Jiaotong University, Xi’an 710049, China; jianzhemg@stu.xjtu.edu.cn (J.M.); ysj2014@stu.xjtu.edu.cn (S.Y.); d.zhao@mail.xjtu.edu.cn (D.Z.); flli@xjtu.edu.cn (F.L.); 2SCNU Environmental Research Institute, Guangdong Provincial Key Laboratory of Chemical Pollution and Environmental Safety & MOE Key Laboratory of Environmental Theoretical Chemistry, South China Normal University, Guangzhou 510006, China; 3School of Environment, South China Normal University, Guangzhou 510006, China

**Keywords:** unidirectional rotation, trajectory surface-hopping simulation, potential energy surface, thermal helix inversion, quantum yield

## Abstract

Working cycle of conventional light-driven molecular rotary motors (LDMRMs), especially Feringa-type motors, usually have four steps, two photoisomerization steps, and two thermal helix inversion (THI) steps. THI steps hinder the ability of the motor to operate at lower temperatures and limit the rotation speed of LDMRMs. A three-stroke LDMRM, 2-(2,7-dimethyl-2,3-dihydro-1*H*-inden-1-ylidene)-1,2-dihydro-3*H*-pyrrol-3-one (DDIY), is proposed, which is capable of completing an unidirectional rotation by two photoisomerization steps and one thermal helix inversion step at room temperature. On the basis of trajectory surface-hopping simulation at the semi-empirical OM2/MRCI level, the *EP→ZP* and *ZP→EM* nonadiabatic photoisomerization dynamics of DDIY were systematically analyzed. Quantum yields of *EP→ZP* and *ZP→EM* photoisomerization of DDIY are ca. 34% and 18%, respectively. Both *EP→ZP* and *ZP→EM* photoisomerization processes occur on an ultrafast time scale (ca. 100–300 fs). This three-stroke LDMRM may stimulate further research for the development of new families of more efficient LDMRMs.

## 1. Introduction

Molecular motors [1,2,3], as one class of molecular machines [4], whose mechanical motion is controlled by an energy source to resist Brownian motion, need an incentive source supply and control system [5]. Light-driven molecular rotary motors (LDMRMs), which can realize unidirectional and repetitive rotation under the stimulation of light energy so as to convert light energy into mechanical energy [5,6], have become more and more stand out because of their green energy resource (light) and ease of control compared to the chemical [7,8,9] and electrical [10,11,12] stimulation motors. For the conventional LDMRMs, especially Feringa-type motors, the motion of rotor with respect to the stator mainly consists of two steps of photoisomerization and two steps of thermal helix inversion (THI) to complete a 360${}^{\circ }$ rotation by introducing a chiral environment [5,6,13,14,15]. The power stroke of LDMRMs is mainly based on the *E→Z* (or *Z→E*) photoisomerization around a central carbon–carbon double bond or carbon–nitrogen double bond [5,6].

The presence of THI steps hinders the rotation of LDMRMs at lower temperatures and also limits the rotation speed and unidirectionality of the LDMRMs [14,16,17,18,19,20]. Therefore, reducing the THI steps is an ongoing research challenge [18,19,20,21,22,23,24]. On the theory side, by introducing a chiral hydrogen bond environment, García-Iriepa et al. [18] proposed a two-stroke photon-only LDMRM, which was predicted to complete the unidirectional rotation with only two photochemical steps (no thermal steps are involved in the mechanism). Based on a bio-inspired 4-hydroxybenzylidene-1,2-dimethylimidazolinone-based molecular photoswitch, a family of two-stroke photon-only LDMRMs were designed by Filatov et al. [19] recently. With the nonadiabatic molecular dynamics (NAMD) simulations on the SSR-BH&HLYP/6-31G(d) level, these two-stroke LDMRMs were predicted to have very high quantum yields of photoisomerization (0.91–0.97) and unidirectionality. A visible-light responsive Schiff-base LDMRM featuring dihydropyridinium and cyclopentenylidene motifs was proposed theoretically by Wang et al. [21] recently, which could rotate unidirectionally with only two photoisomerization processes. Using NAMD simulations with Tully’s fewest switches algorithm [25,26], the quantum yields of almost 70% for each of the two photoisomerizations were predicted. With quantum-classical simulation, a novel molecular motor in which the rotation is caused by the electric coupling of chromophores was suggested by Majumdar et al. [22] very recently, which is driven by light only and can make a full rotation in picoseconds using the power of a single photon.

In experiments, Gerwien et al. [20] proposed a three-stroke photon-only LDMRM which consists of three different isomeric states on the ground state. The unidirectional rotation of one molecular fragment against the other is achieved with up to 98% directionality by the stimulation of visible light. Three new second-generation molecular motors featuring a phosphorus centre in the lower half of the molecule were synthesized by Boursalian et al. [23] recently. These overcrowded alkene-based molecular motors have four diastereomeric states and can interconvert solely photochemically. All-photochemical unidirectional rotation of the new motors was confirmed by kinetic analysis and modelling. A novel three-stage photochemical–thermal–thermal isomerization cycle involving inversion of the configuration of an axial chiral phosphorus stereoelement was also observed by Boursalian et al. [23]. i.e., a three-stroke LDMRM including two THI steps and one photoisomerization step was demonstrated.

During exploring the substituent effect on the oxindole-based LDMRMs synthesized by Roke et al. [27] and Pooler et al. [28] recently, based on electronic structure calculation and nonadiabatic dynamics simulation, we proposed a three-stroke LDMRM, 2-(2,7-dimethyl-2,3-dihydro-1*H*-inden-1-ylidene)-1,2-dihydro-3*H*-pyrrol-3-one (DDIY), including two photoisomerization steps and one thermal helix inversion step. The photoinduced isomerization dynamics of this three-stroke LDMRM was systematically investigated using the trajectory surface-hopping molecular dynamics based on the semi-empirical OM2/MRCI level. The quantum yields of *EP→ZP* and *ZP→EM* photoisomerization processes we obtained are 34% and 18%, respectively. The average lifetimes for the *EP→ZP* and *ZP→EM* processes (about 165 fs and 223 fs, respectively) are both on an ultrafast time scale. The unidirectivity of *EP→ZP* and *ZP→EM* photoisomerization processes are 100% and 96%, respectively.

The paper is organized as follows: In Section 2, we discuss the basic theoretical methods and the simulation details. The results of electronic structure calculation and the nonadiabatic photoisomerization dynamics simulation of *EP→ZP* and *ZP→EM* processes are presented in Section 3. Finally, we summarize our result and discussion in Section 4.

## 2. Computational Details

### 2.1. Density Functional Methods

Geometrical optimization and frequency calculation of DDIY by density functional theory (DFT) were performed with the CAM-B3LYP/6-31G(d) and BH&HLYP/6-31G(d) methods. Taking two dihedral angles as degrees of freedom, the two-dimensional ground state potential energy surfaces (PES) of DDIY were obtained with the relaxed scan method at the CAM-B3LYP/6-31G(d) level. All the DFT calculations were carried out using the GAUSSIAN 09 program [29].

### 2.2. Semiempirical Methods

All semi-empirical calculations were performed with the OM2/MRCI method as implemented in the development version of the MNDO program [30]. This method can balance the computational cost and accuracy well, as shown by many benchmark calculations [31,32,33], and has been used successfully in many calculations of photoinduced processes [34,35,36,37,38,39,40,41,42,43]. The required energies, gradients, and nonadiabatic coupling elements were calculated analytically in the geometry optimizations and dynamics simulations. The self-consistent field (SCF) calculations were performed in the restricted open-shell Hartree Fock (ROHF) formalism, as it provided a better description of the excited-state wave functions. Three reference configurations including the closed-shell ground-state configuration and single and double excitations from the highest occupied molecular orbital (HOMO) to the lowest unoccupied molecular orbital (LUMO) were used to generate multireference configuration interaction (MRCI) expansion.

The active space in the MRCI calculations included eight electrons in eight orbitals (π and π* orbitals). The π-type population (PIPOP) method [41,44] with a threshold of 0.4 was used to identify and trace the π character orbitals and ensure the π orbitals in the active space. Optimizations of the ground state S${}_{0}$ geometries and the first excited-state S${}_{1}$ geometries were performed at the OM2/MRCI level. The Lagrangian–Newton approach [45] was used to locate the S${}_{1}$/S${}_{0}$ minimum-energy conical intersections (CIs) geometries.

The nonadiabatic photoisomerization dynamics of molecular motor DDIY was investigated by the trajectory surface-hopping (TSH) simulations with Tully’s fewest-switches algorithm [25,26,46,47,48]. An empirical decoherence correction (0.1 a.u.) proposed by Granucci et al. [49] was also used. The initial structures and velocities were obtained using the Wigner sampling method [50,51,52]. The nuclear motion was solved using the velocity Verlet algorithm with a constant time step of 0.1 fs, while the time-dependent electronic Schrödinger equation was propagated with a 100 times smaller time step.

## 3. Results and Discussion

### 3.1. Equilibrium Structures

Based on the OM2/MRCI, CAM-B3LYP/6-31G(d), and BH&HLYP/6-31G(d) level of theories, four local minima structures of DDIY on the ground state were obtained. According to the conformation and helicity, these four equilibrium structures are named as *ZP*, *ZM*, *EP* and *EM*, respectively. The definition of helicity took the approach of Karnik et al. [53]. The geometry of the most stable isomer, *ZP*-DDIY, is shown in Figure 1a, while the geometries of the other three isomers are presented in Appendix A. The corresponding geometrical parameters of the four isomers are summarized in Appendix A. As we can see, the optimized geometries obtained from the different theoretical methods above are consistent with each other.

It should be emphasized that the *ZM* isomer of DDIY is quite difficult to locate in the optimization step of all OM2/MRCI, CAM-B3LYP/6-31G(d), and BH&HLYP/6-31G(d) methods. In order to unravel the reason, the potential energy surface near the *ZM* isomer was calculated with a relaxed scan method at the CAM-B3LYP/6-31G(d) level. Taking the two dihedral angles C5-C1-C2-N31 and C2-C1-C5-C23 (labelled according to Figure 1a) as degrees of freedom, the ground state PES near *ZP* and *ZM* isomers is shown in Figure 2a, the corresponding two-dimensional contour of PES is shown in Appendix A. As we can see, the energy barrier from *ZM* to *ZP* estimated from the PES is only about 0.5 kcal/mol. Due to the very low energy barrier from *ZM* to *ZP* isomers, searching for the *ZM* isomer of DDIY is not a trivial task. The thermal isomerization from *ZM* to *ZP* could occur when the ambient temperature is greater than 258 K. Therefore, the photoisomerization starting from the *EP* isomer of DDIY may completely exceed the energy barrier and arrive at the *ZP* thermally stable isomer, without staying at the metastable *ZM* isomer at room temperature. Similarly, taking the two dihedral angles C5-C1-C2-C4 and C2-C1-C5-C23 as degrees of freedom, the ground state PES near *EP* and *EM* isomers obtained with the relaxed scan method at the CAM-B3LYP/6-31G(d) level is shown in Figure 2b, the corresponding two-dimensional contour of PES is shown in Appendix A. As we can see, there is a high energy barrier (about 6.4 kcal/mol) from *EM* to *EP* isomer. So the molecule would stay at the *EM* geometry for a long time and may reach the thermally stable *EP* geometry through the THI process. Thus, we may realize a three-stroke LDMRM, whose working cycle diagram is presented in Figure 1b. At room temperature or even lower, the molecular motor DDIY we proposed may complete a 360${}^{\circ }$ unidirectional rotation by the interconversion of three isomers including two steps of photoisomerization and one step of THI.

Ground state potential energy surfaces of DDIY were also calculated with the relaxed scan method at the OM2/MRCI level, which are in good agreement with the above results obtained with the CAM-B3LYP/6-31G(d) method. The ground state PES near *ZP* and *ZM* isomers as functions of two dihedral angles C5-C1-C2-N31 and C2-C1-C5-C23 calculated at the OM2/MRCI level is presented in Appendix A, the corresponding two-dimensional contour is shown in Appendix A. As we can see, the energy barrier from *ZM* to *ZP* isomer estimated from the PES is about only 0.3 kcal/mol, a little smaller than the value (0.5 kcal/mol) obtained from CAM-B3LYP/6-31G(d) method. The ground state PES near *EP* and *EM* isomers as functions of two dihedral angles C5-C1-C2-C4 and C2-C1-C5-C23 calculated at the OM2/MRCI level is shown in Appendix A, the corresponding two-dimensional contour is shown in Appendix A. As can be seen, there is a high energy barrier from *EM* to *EP* isomer (about 4.0 kcal/mol).

Our above proposition could be supported by the working mechanism of a two-stroke LDMRM DTPN proposed by Filatov et al. [19] recently. With the CAM-B3LYP/6-31G(d) and BH&HLYP/6-31G(d) methods implemented in Gaussian 09, we obtained three ground state equilibrium structures of DTPN, which are presented in Appendix A. The corresponding geometrical parameters of the three isomers are summarized in Appendix A. The ground state PESs of Filatov’s two-stroke LDMRM DTPN near the *EP* and *ZP* isomers were calculated through the relaxed scan method at the CAM-B3LYP/6-31G(d) level, as shown in Appendix A. According to the Appendix A, the energy barrier from the *ZM* to *ZP* isomer is estimated to be only about 0.8 kcal/mol. Thus, molecular motor DTPN may exceed the energy barrier and arrive at a more stable *ZP* isomer at room temperature. Filatov et al. [19] did observe direct *EP→ZP* photoisomerization process of DTPN in their NAMD simulations, while metastable *ZM* geometry was not observed during the working cycle of LDMRM DTPN.

Similar to Filatov’s two-stroke LDMRM DTPN [19], the direct *EP*→*ZP* photoisomerization process of DDIY is expected to occur at room temperature, which may be confirmed by nonadiabatic molecular dynamics simulation. On the basis of trajectory surface-hopping simulation at the semi-empirical OM2/MRCI level, the *EP→ZP* and *ZP→EM* nonadiabatic photoisomerization dynamics of DDIY were systematically studied in the following.

### 3.2. The Nonadiabatic Dynamics of EP→ZP Photoisomerization

Based on the normal vibration modes of the ground state *EP* structure, the initial geometries and velocities of the nonadiabatic dynamics simulation were sampled from the Wigner distribution function. The excited state of S${}_{1}$ corresponds to the single-electron excitation from the HOMO (bonding π orbital) to the LUMO (antibonding π* orbital), with the excitation wavelength at 338 nm. Molecular dynamics simulations of 280 trajectories starting from the S${}_{1}$ excited state were performed with the OM2/MRCI method for 1000 fs. In the 280 trajectories, 275 trajectories reached the ground state within 1000 fs and 95 trajectories underwent *EP*→*ZP* photoisomerization, which means that the quantum yield of *EP→ZP* photoisomerization is estimated to be about 34%.

The average occupation of electronic states S${}_{0}$ and S${}_{1}$ varying with simulation time is shown in Figure 3. The average occupation of the S${}_{1}$ excited state can be approximately fitted with an exponential function: $f\left(t\right)=e^{-a(t-t_{0})}$, where *a* stands for the corresponding decay rate constant and $t_{0}$ is the initial delay time. i.e., the decay mode of *EP→ZP* photoisomerization process is approximately exponential. The fitting curve we obtained is shown in the inset of Figure 3, and the obtained fitting parameters *a* and *t*${}_{0}$ are 0.0087 fs${}^{-1}$ and 50 fs, respectively. Thus, the average S${}_{1}$ excited state lifetime of the *EP* isomer of DDIY is estimated to be about *t*${}_{0}$ + 1/*a* = 165 fs.

Based on all 275 geometries at S${}_{1}$/S${}_{0}$ hopping events, two optimized S${}_{1}$/S${}_{0}$ conical intersections were obtained at the OM2/MRCI level, as shown in Appendix A. Corresponding geometrical parameters of them are summarized in Appendix A. According to the characteristic dihedral angle C5-C1-C2-C4 (55.8${}^{\circ }$ or 111.9${}^{\circ }$), the two CIs are named as *ECI(1)* and *ECI(2)*, respectively. It is clear that both CIs involve obvious pyramidalization at the C2 atom site. Similar pyramidalization of the carbon atom at the stator-axle linkage was also observed in other molecular rotary motors [34,54].

It is helpful for us to understand the decay mechanism through the distribution of geometrical parameters at hopping events. The distributions of C5-C1-C2-C4 and C2-N31-C4-C1 dihedral angles at all 275 S${}_{1}$→S${}_{0}$ hopping events are illustrated in Figure 4, corresponding points of the ground state *EP* isomer, *ECI(1)* and *ECI(2)* are also presented in this figure. As we can see, the most trajectories with successful *EP→ZP* photoisomerization were accessed through hops close to the CIs. Many hops closer to the initial *EP* structure were also observed, almost all of them returned to the reactant. The proportion of the hopping geometries near the initial *EP* geometry is close to that of the two CIs in all trajectories. This explains why the quantum yield of *EP→ZP* photoisomerization is not too high. We should emphasize that all trajectories in our simulations rotated anticlockwise, i.e., the unidirectivity of the *EP→ZP* photoisomerization process is 100%.

In order to understand the *EP→ZP* photoisomerization mechanism of DDIY in detail, time-dependent evolutions of central bond length C1=C2, central dihedral angle C5-C1-C2-C4 and C5-C1-C2-N31, side dihedral angle C2-C1-C5-C23, and pyramid dihedral angle C2-N31-C4-C1 in five typical trajectories (named as trajectory 1–5, respectively) are presented in Appendix A. The corresponding geometrical parameters of the reaction product *ZP* isomer and S${}_{1}$→S${}_{0}$ hopping time are also shown in the figures.

Take trajectory 1 as an example, as shown in Appendix A, after the excitation from S${}_{0}$ to S${}_{1}$, the central C1=C2 double bond is weakened, increasing from its optimized ground state value of 1.37 Å to about 1.45 Å, varying around 1.42 Å until the nonadiabatic decay at 258 fs, after then returning to about 1.37 Å. That is, the excitation from the bonding π orbital of the central C=C bond to the antibonding π* orbital reduces its double bond character obviously. The dihedral angle C5-C1-C2-C4 increased gradually from 9.8${}^{\circ }$ to about 92.9${}^{\circ }$ around 258 fs, after the de-excitation, it increased continually to its optimized ground state value of 181.4${}^{\circ }$ in the *ZP* structure at about 470 fs. The dihedral angle C2-N31-C4-C1, characterizing the pyramidalization at the C2 atom, increased to 38.8${}^{\circ }$ when nonadiabatic decay occurred, after then decreased dramatically to 1${}^{\circ }$ around 300 fs, and varying around 1${}^{\circ }$ until the end of simulation. Both optimized geometries of conical intersection presented in Appendix A and the time dependence of the geometrical parameters shown in Appendix A verify that, after the S${}_{0}$→ S${}_{1}$ excitation, the dynamical process of nonadiabatic decay is followed by twisting about the central C=C double bond and the pyramidalization of the C atom at the stator-axle linkage.

As can be seen in Appendix A, side dihedral angle C2-C1-C5-C23 is the key geometrical parameter to distinguish *ZP* and *ZM* isomers of DDIY. As shown in Appendix A, the dihedral angle C2-C1-C5-C23 vibrated around 28.1${}^{\circ }$ (optimized value in *EP* geometry) until the nonadiabatic decay at 258 fs, after then increased dramatically to 62.7${}^{\circ }$ at about 300 fs, decreased gradually to −31.9${}^{\circ }$ (optimized value in *ZM* geometry) at about 450 fs, i.e., molecular motor arrived at the *ZM* geometry. After staying around the *ZM* geometry for less than 100 fs, the dihedral angle C2-C1-C5-C23 increased gradually to 32.1${}^{\circ }$ (optimized value in *ZP* geometry) at about 625 fs, then vibrated around this value until the end of simulation. Time dependence of geometrical parameters shown in Appendix A verify that, after the S${}_{0}$→S${}_{1}$ excitation of *EP* isomer, molecular motor DDIY arrives at the *ZM* isomer firstly, then reaches the *ZP* isomer in a very short time. i.e., the *EP→ZP* photoisomerization process of DDIY can be realized at room temperature, which confirms our expectation in the beginning.

### 3.3. The Nonadiabatic Dynamics of ZP→EM Photoisomerization

With the same method as above *EP→ZP* photoisomerization nonadiabatic dynamics simulation, nonadiabatic dynamics of *ZP→EM* photoisomerization was systemically investigated. Molecular dynamics simulation of a total of 228 trajectories starting from the S${}_{1}$ excited state (with the excitation wavelength at 340 nm) were carried out for 1000 fs, all trajectories decayed to the ground state before the end of the simulation. In the 228 trajectories, 40 trajectories experienced *ZP→EM* photoisomerization, which means that the quantum yield of *ZP→EM* photoisomerization is estimated to be about 18%.

The average occupation of electronic states S${}_{0}$ and S${}_{1}$ as a function of simulation time is shown in Figure 5. As we can see, the S${}_{1}$ population decay is obviously not exponential. Taking a numerical derivative on the occupation of S${}_{0}$ state over time, as shown in the inset of Figure 5, we can see that the decay mode of the S${}_{1}$ excited state is periodic. Four major hopping event maxima arose at around 130 fs, 350 fs, 585 fs, and 798 fs, respectively. This indicates that the motion of the molecular motor on the PES of S${}_{1}$ excited state towards the conical intersection is regulated by a periodic structural change. The periodic intervals of hopping event maxima in the inset of Figure 5 are roughly in the 213–235 fs range, close to a ground state normal mode of *ZP*-DDIY (142 cm${}^{-1}$, the fifth normal mode, corresponding vibrational duration is 235 fs) involving a swing of phenmethyl ring around the central C=C double bond. Similar periodic decay modes have also been observed in *Z-E* photoisomerization of some azobenzene-based molecules [40,42,43]. From the S${}_{1}$ excited-state lifetimes of all 228 trajectories, the average lifetime of the S${}_{1}$ excited state of the *ZP*-DDIY is estimated to be about 223 fs.

Based on all 228 S${}_{1}$/S${}_{0}$ hopping geometries, two optimized S${}_{1}$/S${}_{0}$ CIs named *ZCI(1)* and *ZCI(2)* were obtained, as shown in Appendix A. Corresponding geometrical parameters of them are summarized in Appendix A. The characteristic dihedral angles C5-C1-C2-C4 of *ZCI(1)* and *ZCI(2)* is −55.0${}^{\circ }$ and −113.5${}^{\circ }$, respectively. Similar to the conical intersection in *EP→ZP* photoisomerization process, both *ZCI(1)* and *ZCI(2)* involve distinct pyramidalization at the C2 atom site.

The distributions of C5-C1-C2-C4 and C2-N31-C4-C1 dihedral angles at all 228 S${}_{1}$→S${}_{0}$ hopping points are illustrated in Figure 6. Corresponding points of the ground state *ZP* isomer, *ZCI(1)* and *ZCI(2)* are also presented in Figure 6. As we can see, the most trajectories that experienced *ZP→EM* photoisomerization were accessed through hops close to the two CIs. A large number of trajectories decayed close to the initial *ZP* structure, almost all of them returned to the reactant *ZP* isomer. That is why the quantum yield of the *ZP→EM* photoisomerization we obtained is a little low. In the 228 trajectories, 218 trajectories rotated anticlockwise, while 10 trajectories decaying close to the initial *ZP* geometry rotated in reverse, which indicates that the unidirectivity of the *ZP→EM* photoisomerization process is about 96%. We should emphasize that all 10 trajectories rotating clockwise in our simulation returned to the reactant. The unidirectivity of the successful *ZP→EM* photoisomerization process is still 100%.

In order to discuss the *ZP→EM* photoisomerization mechanism in detail, time-dependent evolutions of central bond length C1=C2, central dihedral angle C5-C1-C2-N31 and C5-C1-C2-C4, pyramid dihedral angle C2-N31-C4-C1 in five typical trajectories (named trajectory 1–5, respectively) are presented in Appendix A. The corresponding geometrical parameters of the reaction product *EM* isomer and S${}_{1}$→S${}_{0}$ hopping time are also shown in the figures.

Take trajectory 1 as an example, as shown in Appendix A, after the excitation from S${}_{0}$ to S${}_{1}$, the central C1=C2 bond is weakened, increasing from its optimized ground state value of 1.37 Å to about 1.49 Å, varying around 1.42 Å until the nonadiabatic decay at 563 fs, then returning to about 1.37 Å. The dihedral angle C5-C1-C2-C4 increased gradually from −168${}^{\circ }$ to about −90${}^{\circ }$ around 300 fs, varying around −90${}^{\circ }$ until 563 fs, after the S${}_{1}$→S${}_{0}$ nonadiabatic decay, it increased continually to its optimized ground state value of −19.9${}^{\circ }$ in the *EM* structure at about 685 fs. The dihedral angle C2-N31-C4-C1, characterizing the pyramidalization at the C2 atom, kept varying around 5${}^{\circ }$ until 390 fs, then decreased gradually to −31.6${}^{\circ }$ at 563 fs, after then increased to its optimized ground state value of −3.8${}^{\circ }$ around 650 fs. The side dihedral angle C2-C1-C5-C23 decreased from 18.8${}^{\circ }$ gradually to −1.0${}^{\circ }$ at 563 fs, after then decreased continually to −25.5${}^{\circ }$ (optimized ground state value in *EM* isomer) at about 580 fs, varying around −25.5${}^{\circ }$ until the end of 1000 fs. Due to the high energy barrier from *EM* to *EP* isomer, *EM→EP* thermal isomerization was not observed in our simulation. The time dependence of geometrical parameters shown in Appendix A, together with optimized geometries of conical intersections presented in Appendix A, verify that the dynamical process of nonadiabatic decay is followed by twisting about the central C=C double bond and the pyramidalization of the C atom at the stator-axle linkage.

In order to deeply understand the working mechanism of molecular motor DDIY, energy profiles of the S${}_{0}$ and S${}_{1}$ state generated by the linear interpolation among *EP*, *ECI*, *ZM*, *ZP*, *ZCI*, and *EM* structures are shown in Figure 7. As we can see, after the S${}_{0}$→S${}_{1}$ optical excitation of *EP* isomer, molecular motor DDIY rotates around the central C=C double bond in anticlockwise direction, and relaxes rapidly to the S${}_{0}$ state through the conical intersection *ECI*, then it arrives at the metastable *ZM* isomer. Due to the very low energy barrier from *ZM* to *ZP* isomer, as shown in Figure 2a and Figure 7, molecular motor DDIY reaches the ground state *ZP* isomer in a very short time (hundreds of femtoseconds) instead of stopping at the metastable *ZM* isomer.

After the S${}_{0}$→S${}_{1}$ optical excitation, *ZP* isomer in the Franck–Condon region moves in an energy well. Since the left barrier of the energy well is much larger than the right barrier, as shown in Figure 7, most trajectories of *ZP* isomer prefer exceeding the right barrier. i.e., *ZP* isomer of molecular motor DDIY prefers rotating around the central C=C double bond in an anticlockwise direction. After exceeding the right barrier, the *ZP* isomer of molecular motor DDIY relaxes rapidly to conical intersection *ZCI* and decays to the ground state *EM* geometry. The energy barrier from *EM* to *EP* isomer on the ground state, as can be seen in Figure 2b and Figure 7, is more or less compare to the energy barriers of thermal steps in other LDMRMs [28,35,55]. Thus, *EM* to *EP* isomerization through thermal helix inversion on the ground state would subsequently occur. Both nonadiabatic dynamics simulation and reaction energy profiles verify that the molecular motor DDIY can finish a complete 360${}^{\circ }$ rotation by two photoisomerization steps (*EP*→*ZP* and *ZP*→*EM*) and one thermal helix inversion step (*EM*→*EP*) at room temperature.

Although some outstanding developments on photon-only molecular motors [18,19,21,22] have been achieved from a computational perspective, Feringa et al. pointed out very recently that these molecules are often highly challenging to synthesize in experiment [24]. Designing a light-driven molecular motor with fewer operational steps based on synthesized molecular systems may be an effective way. The molecular motor DDIY was designed based on the easy-to-synthesize oxindole-based molecular motors investigated by Roke et al. [27] and Pooler et al. [28] recently, which may reduce the difficulty of synthesis in the experiment. Furthermore, according to our above results, when the energy barrier of thermal helix inversion in conventional four-step light-driven molecular rotary motors becomes small enough, more efficient molecular rotary motors with fewer operational steps may be obtained. To the best of our knowledge, the design of an efficient light-driven molecular rotary motor in this way has not yet been reported in the literature.

## 4. Conclusions

Based on electronic structure calculation at the CAM-B3LYP/6-31G(d), BH&HLYP/6-31G(d), and OM2/MRCI level, together with nonadiabatic molecular dynamics simulation at the OM2/MRCI level, a three-stroke LDMRM, 2-(2,7-dimethyl-2,3-dihydro-1*H*-inden-1-ylidene)-1,2-dihydro-3*H*-pyrrol-3-one (DDIY), is proposed, which is capable of completing an unidirectional rotation by two photoisomerization steps and one thermal helix inversion step at room temperature. The nonadiabatic photoisomerization dynamics of *EP→ZP* and *ZP→EM* of DDIY are systematically investigated by trajectory surface-hopping molecular dynamics at the OM2/MRCI level. The quantum yields of *EP→ZP* and *ZP→EM* processes predicted in our simulations are 34% and 18%, respectively. The decay mode of *EP→ZP* photoisomerization process is approximately exponential, while that of *ZP→EM* photoisomerization process is periodic. The average S${}_{1}$ excited-state lifetime of *EP→ZP* and *ZP→EM* processes are 165 fs and 223 fs, respectively, which indicates that the dynamics of photoisomerization for *EP→ZP* and *ZP→EM* processes are both on an ultrafast time scale. Two optimized conical intersections of *EP→ZP* and *ZP→EM* photoisomerization processes are located. For *EP* and *ZP* isomers of DDIY, after the S${}_{0}$→S${}_{1}$ excitation, the dynamical process of nonadiabatic decay is followed by twisting about the central C=C double bond and the pyramidalization of the C atom at the stator-axle linkage. The unidirectivity of *EP→ZP* and *ZP→EM* photoisomerization processes are about 100% and 96%, respectively. We should emphasize that the *EM→EP* thermal helix inversion in molecular motor DDIY may slow down the unidirectional rotation. Further work to reduce both energy barriers of thermal helix inversion in the conventional four-step light-driven molecular rotary motor is in progress by our group. Our proposed three-stroke LDMRM may stimulate further research for the development of new families of more efficient LDMRMs.

## Figures and Tables

**Figure 1 ijms-23-03908-f001:**
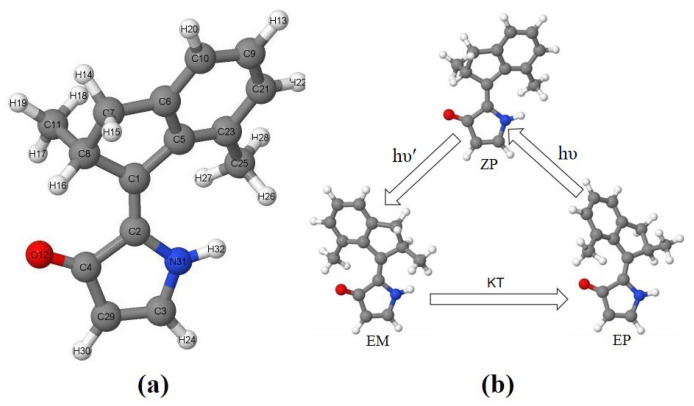
(**a**) Optimized geometry of the *ZP* isomer of DDIY. All atoms are labelled. (**b**) The schematic diagram of a working cycle of the three-step LDMRM DDIY. All geometries are optimized at the OM2/MRCI level.

**Figure 2 ijms-23-03908-f002:**
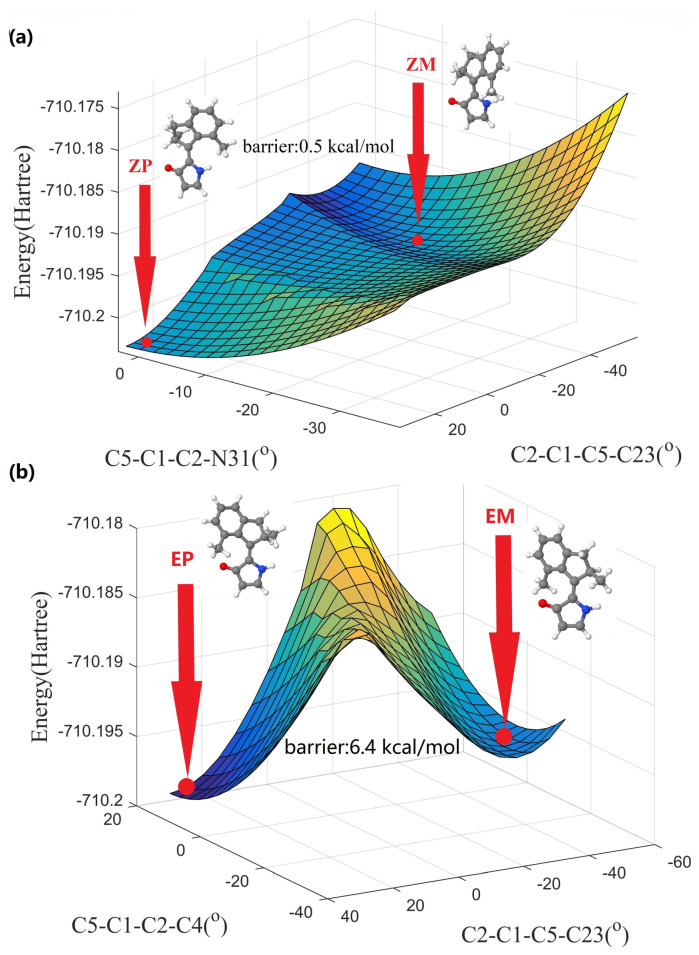
(**a**) Ground state potential energy surface near *ZP* and *ZM* geometries corresponding to dihedral angles C5-C1-C2-N31 and C2-C1-C5-C23. (**b**) Ground state potential energy surface near *EP* and *EM* geometries corresponding to dihedral angles C5-C1-C2-C4 and C2-C1-C5-C23. The above PESs were obtained with relaxed scan method at the CAM-B3LYP/6-31G(d) level, as implemented in Gaussian 09 program [29]. The energy barrier from *ZM* to *ZP* isomer, together with energy barrier from *EM* to *EP* isomer, are shown in subfigures (**a**,**b**), respectively.

**Figure 3 ijms-23-03908-f003:**
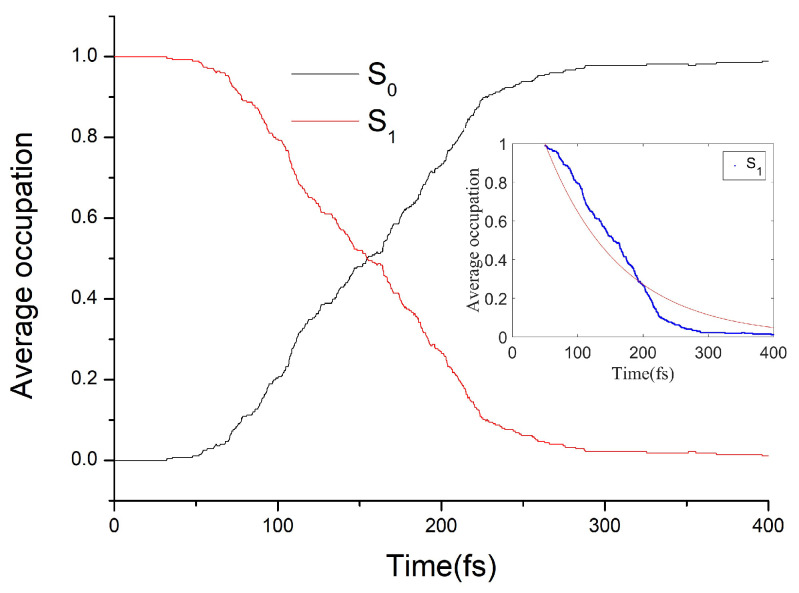
Average occupation of the electronic states S${}_{0}$ and S${}_{1}$ as a function of simulation time. The occupation of the S${}_{1}$ excited state over time fitted by the exponential function is shown in the inset.

**Figure 4 ijms-23-03908-f004:**
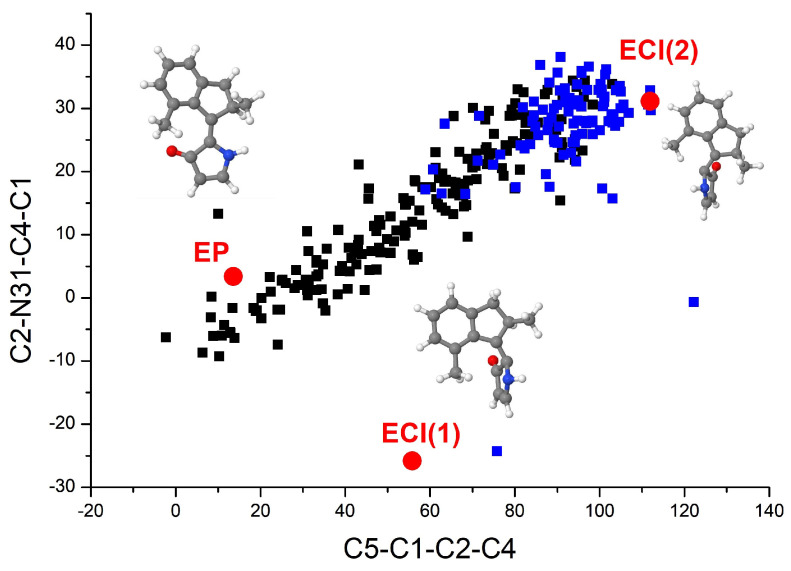
Distribution of the C5-C1-C2-C4 and C2-N31-C4-C1 dihedral angles at the hopping events of 275 trajectories starting from the *EP* structure of DDIY. Black squares denote trajectories returned to the reactant *EP* isomer, while blue squares denote trajectories experienced *EP→ZP* photoisomerization. The ground state *EP* isomers, *ECI(1)* and *ECI(2)* are also presented in this figure.

**Figure 5 ijms-23-03908-f005:**
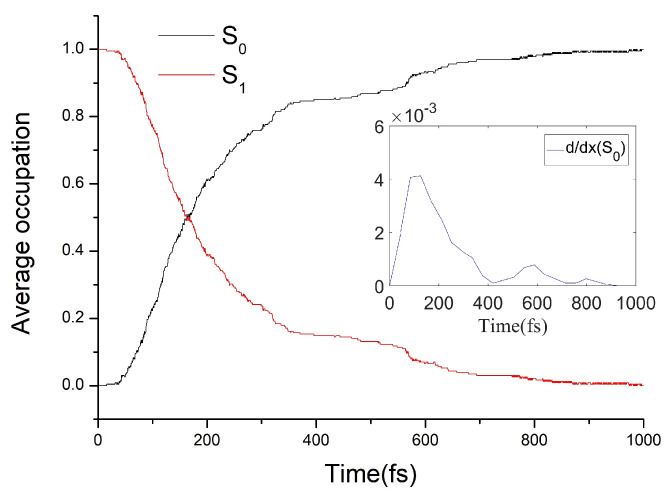
Average occupation of the electronic states S${}_{0}$ and S${}_{1}$ as a function of simulation time. Population decay with the derivative of the S${}_{0}$ population is shown in the inset.

**Figure 6 ijms-23-03908-f006:**
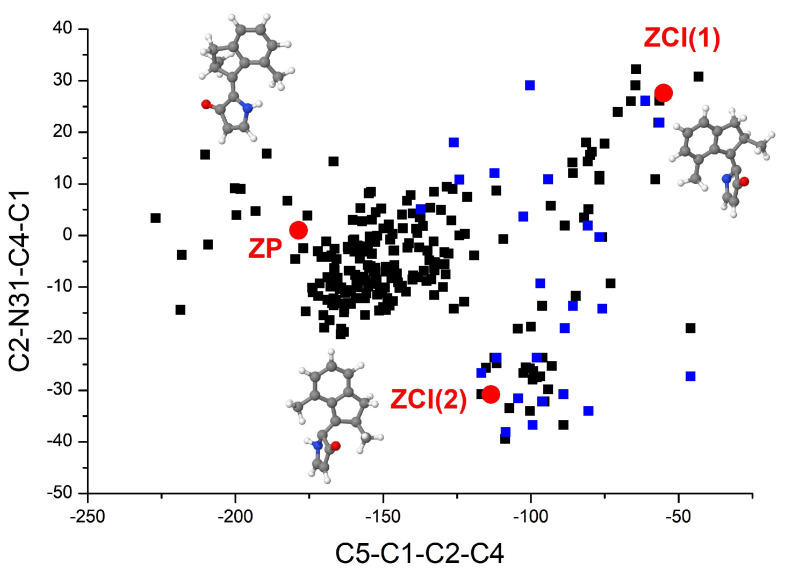
Distribution of the C5-C1-C2-C4 and C2-N31-C4-C1 dihedral angles at the hopping points of all 228 trajectories starting from the *ZP* structure of DDIY. Black squares denote trajectories returned to the reactant *ZP* isomer, while blue squares denote trajectories experienced *ZP→EM* photoisomerization. The ground state *ZP* isomers, *ZCI(1)* and *ZCI(2)* are also presented in this figure.

**Figure 7 ijms-23-03908-f007:**
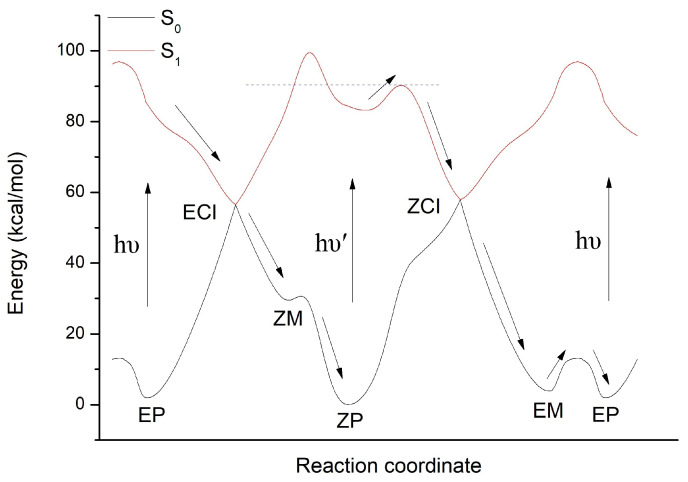
OM2/MRCI energy profiles of the S${}_{0}$ and S${}_{1}$ state generated by the linear interpolation among *EP*, *ECI*, *ZM*, *ZP*, *ZCI*, and *EM* structures of DDIY. The relative energies are calculated with respect to the ground state *ZP* conformation. The black arrows indicate the reaction pathway.

## Data Availability

Not applicable.

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
