# Peer review of "Design and Nonadiabatic Photoisomerization Dynamics Study of a Three-Stroke Light-Driven Molecular Rotary Motor"

_ijms, 2022, doi:10.3390/ijms23073908_

Round 1

Reviewer 1 Report

A nice paper discussing several modifications to standard design of light driven motors that may provide insight into how motor operation can be improved.

Reviewer 2 Report

The Authors focused on the working cycle of conventional light-driven molecular rotary motors (LDMRMs), especial Feringa-type motors, which usually have four steps, two photoisomerization steps and two thermal helix inversion (THI) steps. The Authors proposed a three-stroke LDMRM, 2-(2,7-dimethyl-2,3-dihydro-1H- inden-1-ylidene)-1,2-dihydro-3H-pyrrol-3-one (DDIY), which is capable of completing an unidirectional rotation by two photoisomerization steps and one thermal helix inversion step at room temperature. They analysed the mentioned problem using semi-empirical OM2/MRCI level. The article seems to be interesting for a large number of scholars and engineers. It creates a logical semi-empirical research and that is why in my opinion could be published in "International Journal of Molecular Sciences". Some of the comments on the manuscript are listed below.

1) Line 13 and 14; some keywords have been already used in the title of the manuscript. Please change them into different ones (to avoid the keywords repetition with the words used in the title).

2) Please consider adding some more explanations or a drawing concerning the mechanism of rotational movement of your motor.

3) For the Reviewer it is not quite clear what is new in this manuscript in comparison with the published earlier Authors’ articles?

Reviewer 3 Report

The authors propose a new chemical substance for light-driven molecular motor applications.  A complete rotation of the molecule involves three steps; two light-induced isomerisations and one thermal helix inversion. The different steps are followed through semiempirical structure calculations and molecular dynamic simulations.  The paper is  transparent, the results are convincing; it certainly deserves publication. Nevertheless, from the discussion it is missing a comparison with the substances cited e.g. in Ref 20-22 of the paper, where no thermal step is involved in the cycle. In the case discussed in the present work, although the light-induced transformation undergo in sub-picosecond time intervals, the thermal conversion slows down the process considerably. Does this have any advantage? 

Reviewer 4 Report

In this manuscript, the authors propose a three-stroke light-driven molecular rotary motor (LDMRM) that is capable of completing an unidirectional rotation by two photoisomerization steps and one thermal helix inversion step. These results can stimulate further research for the development of new families of efficient LDMRM. I would like to recommend its publication if the authors can clarify the following concerns:

  1. The novelty of this study is not clear. The authors need to list a Table to compare the main findings between this manuscript and previous studies.
  2. The level of English throughout the manuscript needs to be improved. I would suggest the authors to check the manuscript and refine the language carefully.
